# Impact of vitamin D status and cathelicidin antimicrobial peptide on adults with active pulmonary TB globally: A systematic review and meta-analysis

**Ester Lilian Acen**[1]*, **Irene Andia Biraro**[2,3], **William Worodria**[4], **Moses L. Joloba**[5], **Bill Nkeeto**[6], **Joseph Musaazi**[7], **David Patrick Kateete**[5]

**1** Department of Physiology, School of Biomedical Sciences, College of Health Sciences, Makerere University, Kampala, Uganda, **2** Department of Internal Medicine, School of Medicine, College of Health Sciences Unit Makerere University, Kampala, Uganda, **3** Medical Research Council/Uganda Virus Research Institute and London School of Hygiene and Tropical Medicine Uganda Research Unit, Entebbe, Uganda, **4** Pulmonary Division, Department of Internal Medicine, Mulago National Referral Hospital, Kampala, Uganda, **5** Department of Immunology and Molecular Biology, School of Biomedical Sciences, College of Health Sciences, Makerere University Kampala, Uganda, **6** Department of Policy and Development Economics, School of Economics, College of Business and Management Sciences Makerere University, Kampala, Uganda, **7** Infectious Diseases Institute, College of Health Sciences, Makerere University, Kampala, Uganda

* mulamester82@gmail.com

**Data Availability Statement:** All data are contained within the paper and supporting information files.

## Abstract

### Background

Tuberculosis remains a global threat and a public health problem that has eluded attempts to eradicate it. Low vitamin D levels have been identified as a risk factor for tuberculosis infection and disease. The human cathelicidin LL-37 has both antimicrobial and immunomodulatory properties and is dependent on vitamin D status. This systematic review attempts to compare vitamin D and LL-37 levels among adult pulmonary tuberculosis patients to non-pulmonary TB individuals between 16–75 years globally and to determine the association between vitamin D and cathelicidin and any contributing factor among the two study groups.

### Methods/Design

We performed a search, through PubMed, HINARI, Google Scholar, EBSCOhost, and databases. A narrative synthesis through evaluation of vitamin D and LL-37 levels, the association of vitamin D and LL-37, and other variables in individual primary studies were performed. A random-effect model was performed and weighted means were pooled at a 95% confidence interval. This protocol is registered under the International Prospective Register of Systematic Reviews (PROSPERO), registration number CRD42019127232.

### Results

Of the 2507 articles selected 12 studies were eligible for the systematic review and of these only nine were included in the meta-analysis for vitamin D levels and six for LL-37 levels.

**Funding:** This work has been funded by The Africa Center of Excellence in Materials, Product Development and Nanotechnology (MAPRONANO ACE) Makerere University.

**Competing interests:** The authors have declared that no competing interest exist.

Eight studies were performed in Asia, three in Europe, and only one study in Africa. The mean age of the participants was 37.3±9.9 yrs. We found low vitamin D and high cathelicidin levels among the tuberculosis patients compared to non-tuberculosis individuals to non-tuberculosis. A significant difference was observed in both vitamin D and LL-37 levels among tuberculosis patients and non-tuberculosis individuals (p = < 0.001).

## Conclusion

This study demonstrated that active pulmonary tuberculosis disease is associated with hypovitaminosis D and elevated circulatory cathelicidin levels with low local LL-37 expression. This confirms that vitamin D status has a protective role against tuberculosis disease.

## Introduction

Tuberculosis (TB) caused by *Mycobacterium tuberculosis(Mtb)*remains a global threat and a public health problem [1,2]. According to the 2018 World Health Organization (WHO) report, an estimated 10 million people globally developed the disease that year and accounted for about 1.3 million deaths that occurred in the same year [3,4]. Innate immunity and other host genetic factors are involved in the progression of TB infection to TB disease, and vitamin D is one known risk factor [5]. Worldwide research suggests that vitamin D deficiency (VDD) is found at all ages and has been linked to infectious and non-infectious diseases [5–8]. Moreover, vitamin D3 may be obtained from sunshine exposure and vitamin D2 form is acquired from the diet and eventually synthesized and converted to immunomodulatory 1, 25 Dihydroxy vitamin D3 (1, 25(OH) D3) [6,9–11]. Nonetheless, the widely accepted indicator for vitamin D status is 25(OH) D [12–14].The involvement of vitamin D in protection against TB is demonstrated by the modulation of both innate and adaptive immune responses [15–17]. Sufficient vitamin D levels enhance the expression of LL-37antimicrobial peptide in the macrophages leading to the destruction of *Mtb* and consequently autophagy [4,18,19].VDD is a known risk factor for alterations in the immunomodulatory effects of vitamin D and LL-37, and demonstration of its role in TB pathogenesis has been achieved by in-vitro studies [4,5]. Additionally, studies have reported low vitamin D levels among TB patients compared to their healthy contacts [20–26]. One such in-vitro study was performed by Rook *et al* in which a decline in the growth of *Mtb* was noted in the macrophages incubated with 1, 25(OH)2D3 [5]. This decline in mycobacterial load was further confirmed by another study that demonstrated macrophages releasing LL-37 in the presence of 1,25(OH)2D [27,28]. Invading pathogens including *Mtb* are destroyed by LL-37, an antimicrobial peptide through the direct killing and destruction of the bacterial cell membrane integrity, counteraction of bacterial toxins by neutralizing them and forming a biofilm, and consequently inducing autophagy [15,28–30].LL-37 is regulated through the toll-like receptor (TLR) pathway and is dependent on the amount of vitamin D [15,31].Insufficient 25(OH) D levels prevent the induction of the antimicrobial peptide(CAMP) gene leading to inadequate LL-37 expression [31]. LL-37 has both antimicrobial and immunomodulatory properties and can be detected in blood circulation and expression in cells through mRNA [15]. Its immunomodulatory properties are demonstrated by both pro- and anti-inflammatory activities in TB infection [32]. However several studies globally have reported inconclusive results with regards to the association of vitamin D status and LL-37in TB disease [23,33,34]. Systematic reviews and meta-analysis studies have previously been performed to further examine the relationship between vitamin D and TB to harmonize the

inconsistencies found in primary studies [12,17,35]. Consequently for better understanding of host directed therapy in TB, there is need to further understand the protective relationship between vitamin D and LL-37.No systematic review has previously been performed to date to describe the evidence of this relationship. The objective of our systematic review was to compare vitamin D and LL-37 levels among adult pulmonary TB (PTB) patients to non-pulmonary TB adults between 16–75 years globally and to determine the association between vitamin D and LL-37 and any contributing factor among the two study groups. This synthesized and harmonized evidence on the relationship between the two molecules may identify gaps in the literature of TB pathogenesis and consequently improve policy, practice and research with regard to TB preventive strategies.

## Materials and methods

### Study design, inclusion and exclusion criteria

This protocol was approved by Makerere University School of Biomedical Sciences Higher Degrees Research and Ethics Committee (#SBS-637 and the National council of Science and Technology (HS2639). This systematic review and meta-analysis were performed according to the Preferred Reporting Items for Systematic Reviews and Meta-Analyses (PRISMA)2009 checklist (see S1 File) following a protocol set a priori. The systematic review protocol was registered under the International Prospective Register of Systematic Reviews (PROS-PERO), registration number CRD42019127232 https://www.crd.york.ac.uk/PROSPERO. (see S2 File). Standard methods expected by Cochrane systematic review guidelines were followed for the search strategy, data collection, publication, and risk of bias assessment and statistical analysis. Two reviewers independently used PECO criteria to review the inclusion and exclusion criteria of studies. The systematic review population was active pulmonary TB patients or pulmonary TB patients, vitamin D status was the exposure, the comparator group were the non- Pulmonary TB individuals or Normal controls (NC) and the primary outcome was LL-37 expression. The secondary outcome was the association between vitamin D status and LL-37 levels. Primary studies reporting on vitamin D levels or VDD in adult active pulmonary TB patients with their Normal controls between the ages of 16–75 years were selected. For this review, we defined levels less than 20 ng/ml (<20 ng/ml) as vitamin D deficient and 21–29 ng/ml as vitamin D insufficient or hypovitaminosis D and sufficient vitamin D was equal to or above 30 ng/ml, (≥ 30 ng/ml). There was no restriction on gender inclusion in this review. However, only case-control and comparative cross-sectional studies were included in this review. All studies highlighting information on the association, correlation, relationship, impact, or effect of vitamin D status on LL-37 expression were included. There was no restriction on language and the previous year of publication until the end of the search date. Non-English articles were translated using Google translate. However, we excluded studies reporting on vitamin D and LL-37 among children, those with a population of TB, and household contacts only with no Normal controls. Studies that reported on vitamin D or LL-37 levels only in TB disease were also excluded. Further still, articles that reported on vitamin D and LL-37 levels in other patients' populations other than pulmonary TB were not included. Nonetheless, we included studies that reported on vitamin D and LL-37 levels in a population of pulmonary TB combined with another disease. We further excluded articles on cells and animal models, case reports/ series, and clinical trials.

### Search strategy and review of studies

A search was performed, through PubMed, HINARI, Google scholar, and EBSCOhostdatabases. Primary studies addressing vitamin D status and LL-37 levels in pulmonary TB patients

and their controls globally were included. A search stream was developed using the set search terms and Boolean words such as "AND" when adding new search terms and "OR" when adding to similar terms. The first search began in May 2019 through PubMed search engine using the terms Vitamin D, OR "25(OH) D" OR "cholecalciferol" OR "Hypovitaminosis D",OR "vitamin D deficiency"; AND "cathelicidin" OR "LL-37" OR "antimicrobial peptide" AND "Tuberculosis", OR "TB" OR "Pulmonary TB" OR "*Mtb*". The same search was performed in other databases mentioned earlier. Dissertation and thesis grey data were searched for in the proQUEST database, world health Organization (WHO) websites, and vitamin D conference proceedings. An additional search through reference lists of retrieved articles was performed to identify other potential articles. The final search was performed in July 2019 then articles were sent to the endnote reference manager. The above search was performed independently by two reviewers (EA and BN). This was done byscoping through citations using titles and abstracts without blinding to capture relevant studies. Further screening for full text was performed to includeonly relevant articles. Any discrepancies between the two reviewers were resolved by the third reviewer (DK) who acted as a tie-breaker. All articles searched by the two reviewers were combined in the Endnote X7 reference manager by the main reviewer. Our Endnote reference was then checked and all duplicates were removed. The PRISMA 2009checklist was used to direct the review reporting process. Similarly, the PRISMA flow diagram (Fig 1) was constructed to indicate the selection and outcome of articles that were reviewed according to the inclusion and exclusion criteria set.

## Data extraction for systematic review

Data were extracted independently by the two reviewers EA and BN which was then harmonized into a standard excel datasheet. The following data were extracted from each selected study: title, first author, year of publication, country, latitude, study type, sample size, population, TB cases, controls, age; laboratory test used to confirm TB; type of sample analyzed; predictor variables; percentages of male and female serum cut-offs, association /correlation of LL-37 and vitamin D levels, or relationship of low vitamin D levels and LL-37 expression response.

## Data management and statistical analysis

A narrative synthesis through evaluation of vitamin D, LL-37 levels, association of vitamin D and LL-37, and other variables in individual primary studies was performed. Data from the final studies were summarised in an excel spreadsheet in 2007 and analysis was performed using STATA (Stat Corp. STATA15.0, College Station, Texas USA. The meta-analysis was performed using the random effect model by pooling standard mean differences (SMD) of vitamin D and LL-37 levels among TB and non- TB individualsat a 95% confidence interval(CI) with αset at 0.05%. Heterogeneity Chi-square test was used to assess the presence of variation among the selected studies and the level of variation was detected by calculation of $I^2$ using percentages 25% represents low 50% moderate and 75% high heterogeneity respectively. Forest plots were presented for visual display of variation. Funnel plots of SMD and standard error (SE) were used to identify publication bias. Other methods that were used to assess publication bias were Begg's test and Egger's bias test.

## Publication bias and risk of bias assessment

The Grading of Recommendations and Assessment Development and Evaluation (GRADE) approach was utilized to assess the quality of evidence [36]. Five items including the risk of bias, imprecision by assessing 95% confidence interval, inconsistency by finding out if studies

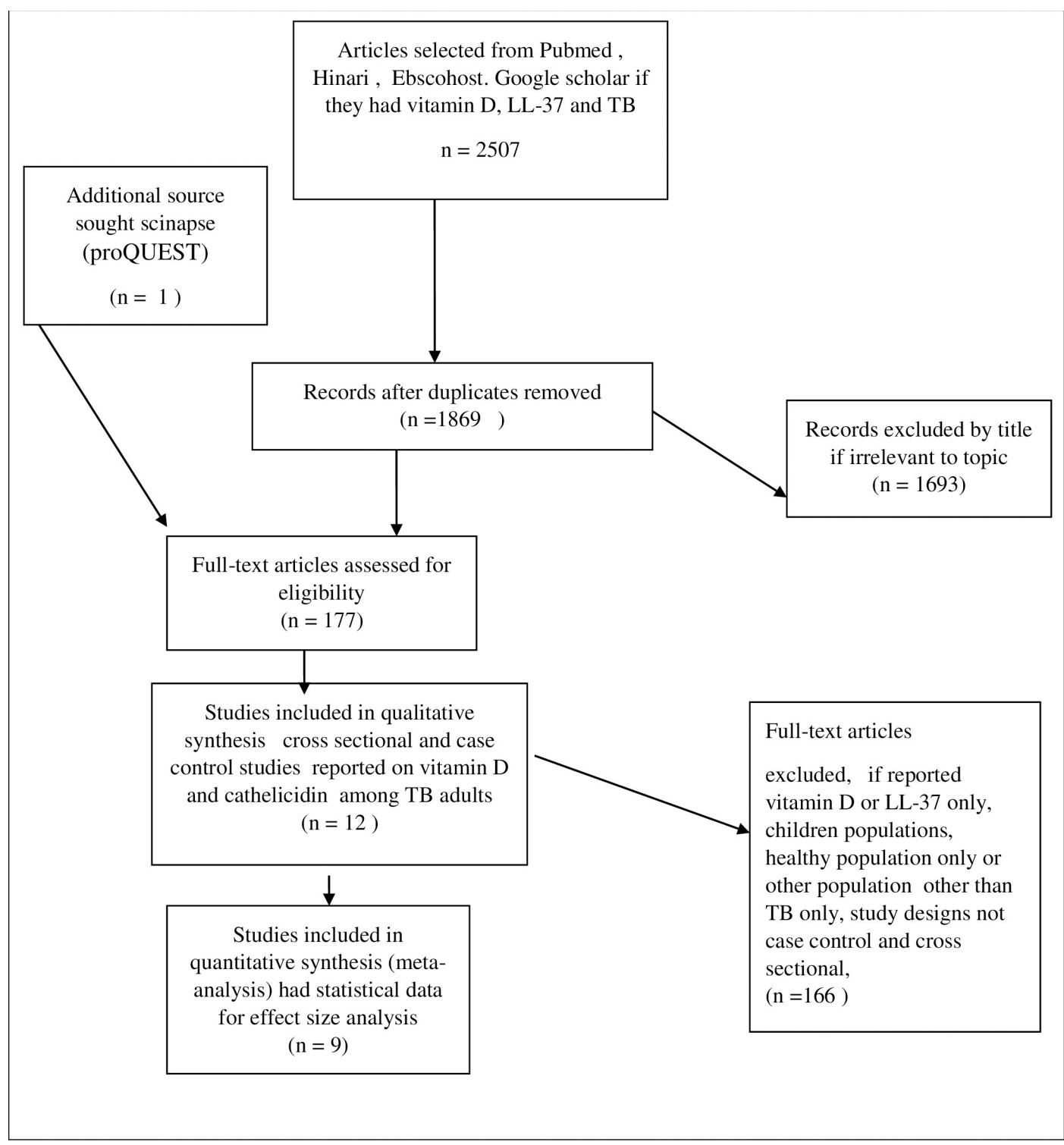

**Fig 1. Flow diagram of review searching and selection process.**

have consistent effects, indirectness by assessing if studies have comparable exposure in the population studied, and publication bias was assessed by making inferences of missing data and exploring the statistics and methods used. The funnel plots method was performed to

assess the visual symmetry of review studies, where the effect size of SMD was measured against the precision SE of vitamin D and LL-37 levels. Begg's ranks correlation analysis test for small-study effects was performed on the effect size against SE of vitamin D and LL-37 levels of TB patients and controls and Kendall score was determined. In the Egger's test regression analysis was performed on the effect size estimate against its SE of the two molecules in TB patients and controls. Asymmetry was measured using the intercept line. The AMSTAR 2 tool was used to measure methodological and statistical quality. The risk of bias was assessed by screening raw data to check for completeness of the information and any other anomalies included by the two reviewers. The quality of the primary studies included in this systematic review was paramount therefore studies with reasonably valid association results were included. The selection of studies was performed by comparing methodological procedures including the study design, type of sample, laboratory analysis, the clinical condition being pulmonary TB, and statistical analysis, including controls, means, and SD or median and interquartile range.

## Results

### Search results

A total of 2507 articles were selected according to the search terms through PubMed, HINARI Ebscohost, and Google scholar. An additional source was sought from which a thesis on the topic was found giving a total of 2508 articles. From these 639 duplicates were removed. After further scrutiny by abstract, 1693unrelated articles were excluded leaving a total of 177 that were eligible for full article assessment. Of these, 11 articles [5,15,33,37–44] were included in the systematic review and only nine were included in the meta-analysis. Fig 1 shows the search strategy and selection process of the relevant articles that were included in this review.

### Study characteristics

Of the 12 studies selected, eight of these were performed in Asia, three from Europe, and only one study from the African continent. The majority of the studies were from India. Table 1 shows the characteristics of the studies selected. Eighty-two percent of the included studies were case-control studies and the rest were cross-sectional studies. Six of these studies had a small sample size of fewer than 100 participants while the highest sample size had461 participants. The mean age of the review studies among the TB patients was37.3±9.9 yrs. Ten studies detected the presence of *Mtb* using a sputum smear test. Of these only two studies confirmed TB using culture methods and another confirmed using GeneXpert MTB/RIF. However, two studies in the review did not report on the method with which *Mtb* was detected. Variable methods were used in the estimation of vitamin D and LL-37 levels. For vitamin D estimation, five studies used the Enzyme-linked immunosorbent assay (ELISA)method, one study used the Tandem mass spectrometry (TMS) method, two studies the Diasorin assay and two studies the chemiluminescent method. Cultures stimulated with *Mtb* in the presence of vitamin D were used to determine the expression of vitamin D receptor(VDR)(messenger RNA)mRNA by the other studies. Equally, for LL-37 expression, the ELISA method was predominantly used by eight studies, two used Immunohistochemistry, one study real-time Polymerase Chain Reaction(PCR), and one study that did not report the method of estimation they used. There was heterogeneous reporting of variables with different detection limits and units of measure. Some studies presented results using means and SD while others reported median and inter-quartile ranges. Six studies used serum samples for the determination of vitamin D levels, three used plasma samples, one study used Peripheral blood mononuclear cells (PBMCs) and three of the studies did not report on the type of samples that were used. Concerning LL-37

**Table 1. Characteristics of reviewed studies.**

| No | Author | Country | Year | Study design | Population | size | Case | control | Site | Diagnosis tool | Sample type | methods | Vit D ng/ml PTB | Vit D ng/ml NC | Association | LL-37 ng/ml/fold PTB | LL-37 ng/ml/fold NC | Association |
|---|---|---|---|---|---|---|---|---|---|---|---|---|---|---|---|---|---|---|
| 1 | Zhan and Jiang | China | 2015 | Cross-sectional | PTB/volunteers | 60 | 30 | 30 | Inpatient chest hospital | smear-positive | Serum/plasma | TMS/ELISA | 12.04 | 17.44 | Low in PTB than NC | 44.5 | 32.3 | High in TB low in NC |
| 2 | K. Majewski et al | Poland | 2018 | case-control | PTB/random controlled study | 107 | 46 | 61 | Pulmonary Disease Hospital | Tuberculin Skin Test/culture | plasma | ELISA | 24.18 | 26.1 | no difference in PTB and NC | 7.45 | 1.41 | High in TB low in NC |
| 3 | K. Afsal et al | India | 2014 | case-control | PTB/students/staff university | 43 | 20 | 23 | Clinics from institutes | smear-positive | PBMCs/culture supernatants | Culture/ELISA | 3.08 | 4.01 | Not mentioned | 3.6 | 4.5 | High in TB and NC |
| 4 | Iqbal et al | Pakistan | 2015 | case-control | PTB/ Administration and lab staff | 39 | 22 | 17 | Out-patients clinic | smear-positive | plasma | ELISA | 15.0 | 15.5 | low in PTB and NC | 42.7 | 29.4 | High in TB low in NC |
| 5 | I.V. Belyaeva et al | Russia | 2017 | case-control | PTB \healthy donors | 96 | 55 | 41 | St Petersburg Research Institute/Sanatorium | smear/culture | Serum/plasma | ELISA | 13.2 | 19.3 | Low both PTB and NC | 50.1 | 39.8 | High in TB and NC |
| 6 | E. korucu et al | Turkey | 2014 | case-control | Not mentioned | 50 | 30 | 20 | Not mentioned | culture | serum | Not mentioned | 22.5 | 33.7 | Low in PTB | 0.056 | 0.1277 | Low in PTB |
| 7 | S.Panda et al | India | 2019 | cross-sectional | PTB/ general population | 150 | 80 | 70 | DOTs centres | smear/genexpert | serum | Chemiluminescence/ELISA | 11.6 | 21.5 | low in PTB compared to NC | 8.83 | 3.75 | High in TB low in NC |
| 8 | S.Panda et al | India | 2019 | Cross sectional | PTB/ general population | 300 | 150 | 150 | DOTs centers | smear/gen expert | serum | Chemiluminescence/ELISA | 13.28 | 23.31 | low in PTB compared to NC | 10.36 | 5.99 | High in TB low in NC |
| 9 | S. Ashenafi et al | Ethiopia | 2018 | case-control | TB/Non-TB Ethiopian Swedish | 217 | 140 |  | Chest clinic University Hospital/korilinsk | smear-positive | Plasma/pbmcs | DiaSorin assay ELISA | 30 | 11.6 | low in PTB and black NC compared to Swedish NC | 4.0 fold | 0.5fold | High in TB low in NC |
| 10 | S.Rahman et al | Russia | 2015 | Case control | TB/University staff | 29 | 19 | 10 | Surgical department in the hospital | smear-positive | Serum/Lung biopsy | DiaSorin assay Histochemistry | 15.1 | 26.0 | low in PTB compared to NC | 5fold | 9fold | Low in TB low in NC |
| 11 | P. Selvaraj et al | India | 2009 | case-control | PTB /Staff and trainees of TB research center | 125 | 65 | 60 | Institute of thoracic medicine and thoracic hosp | smear/culture | Plasma/culture | ELISA/PCR | 0.1 | 0.0641 | high in PTB compared to NC | 8.0 fold | 6.0 fold | High In TB Low in |
| 10 | B. Kiran | India | 2017 | Case-control | TB/University staff | 461 | 280 | 181 | Pediatrics and pulmonology department SRM hospital | Smear/Tuberculin Skin Test/X-ray | Serum | Chemiluminescence/ELISA | 21.29 | 16.56 | Low in PTB and NC | 3.3 | 3.2 | Low in TB and NC |

levels measurement, three studies used serum samples, three used plasma, one lung biopsies, two cultures supernatants, one PBMC samples, and two studies did not report the type of sample used. Vitamin D deficiency predictor variables were reported by only three studies and these comprised of environment, genetics, season variation nutrition, lifestyle, and immunological factors. Among the 12 studies we reviewed, only one study recruited HIV-positive individuals. Five studies excluded HIV patients, five did not state if HIV screening was performed, and one study did not screen for HIV.

## Low vitamin D levels among TB patients

Of the 12 studies included in our systematic review, ten studies measured 25(OH)D levels to determine vitamin D status and one study analyzed1,25(OH)2 D3, Levels. Inconsistency was observed in the definition of vitamin D deficiency among the ten studies and four studies did not define vitamin D status. One study defined vitamin D deficiency as < 25 nmol/L (10.0 ng/ml). In this review we defined vitamin D status according to the Endocrine Society; (deficiency 20 ng/ml, insufficiency 21–29 ng/ml, sufficiency >30 ng/ml, toxicity >150 ng/ml), given that half of our studies used this definition. Accordingly, results of vitamin D deficiency estimated by 25(OH) D levels from nine studies reported low levels among the TB patients. Six studies reported vitamin D deficiency levels of > 20 ng/ml [5,11,15,29,33,39] and three reported insufficient levels between 21–29 ng/ml [38,41,42]. Conversely, severe vitamin D deficiency as low as >10 ng/ml was reported by a study from Russia [43]. The study that measured 1,25 Dihydroxy Vitamin D3, found high levels (2 fold relative change of mRNA expression) among TB patients, (90 pg/ml in the cultured cells treated with vitamin D3) [44]. The last study found a higher expression of vitamin D receptor (VDR) mRNA in stimulated culture cells treated with 1,25 Dihydroxy Vitamin D3 compared to those without (p = 0.024) [37]. Among the control group, six studies registered deficient vitamin D levels, and noteworthy is that two of these found lower levels compared to their TB patients [15,42]. Three studies found insufficient vitamin D levels [5,41,43] and one study reported sufficient vitamin D levels in their control group [38]. The expression of mRNA was lower in the control group in the two studies previously mentioned above. When vitamin D levels were compared between TB patients and their controls, 75% (9) of the studies reported a significant difference. Two studies reported no significant difference between the two groups and one study that had Ethiopian and Swedish controls, found a significant difference between the TB patients and the Swedish controls (p <0.001), and no difference was observed among the TB patients and Ethiopians non TB controls. One study had vitamin D levels lower in the controls than TB patients however they did not report the strength of association. Regarding HIV status, five studies excluded HIV Positive patients, one study did not test for HIV four, and studies did not mention if HIV testing was done. We also found one study that reported higher vitamin D levels among TB/HIV co-infected and HIV patients compared to their HIV-negative counterparts [15].

## Meta-analysis of vitamin D levels in TB

A meta-analysis was performed with data from nine eligible studies using a random-effect model where the SMD was pooled because of the different detection limits and units of measure in the primary studies. Fig 2 shows a forest plot of studies that were included in the meta-analysis with details of estimates SMD of vitamin D levels from each study and the overall pooled effect,(SMD -0.625,95% CI (-0.784–0.466),P < 0.001. The weights in % show the influence of each primary study on the pooled effect using sample size and CI. Lower vitamin D levels were observed among the TB patients compared to the non TB individuals with a significant difference, P < 0.01 as noted by the diamond on the left side of the no-effect line.

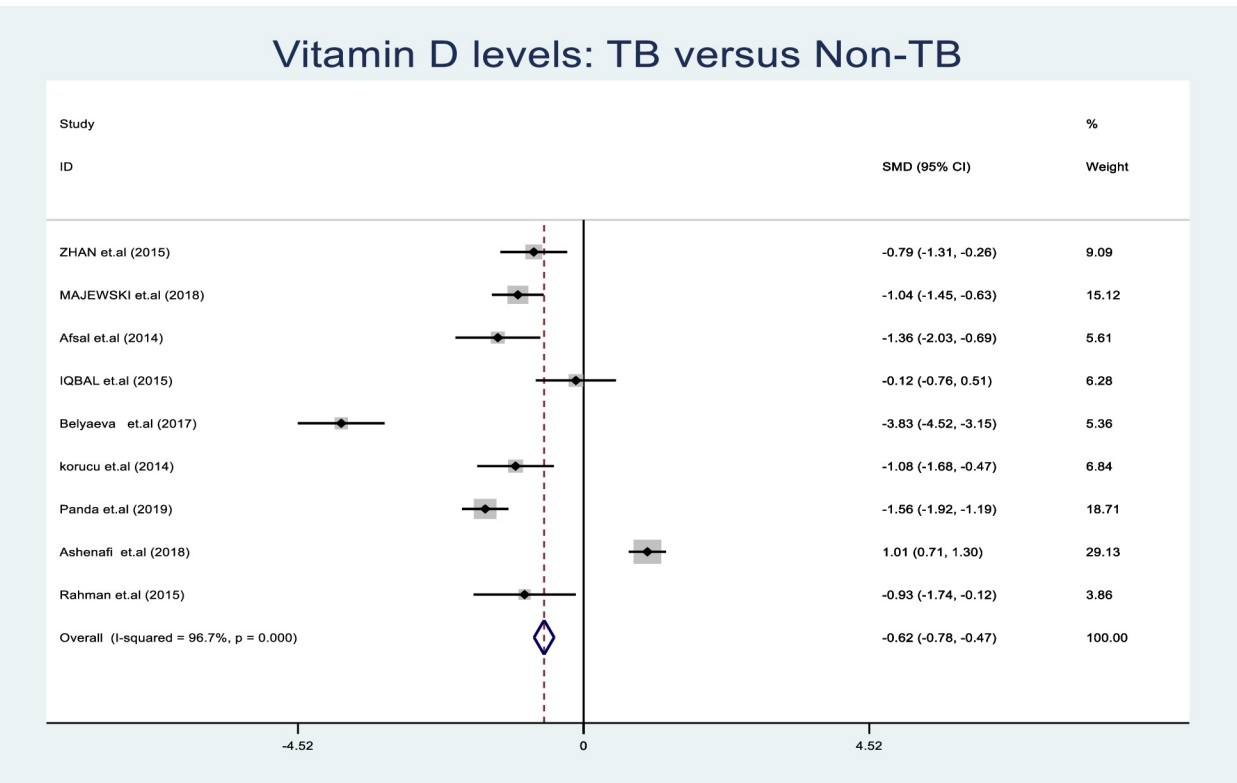

**Fig 2. Forest plot showing the comparison of vitamin D levels among TB patients and non TB individuals: Overall effect of continuous variables using the random effect model.** The solid vertical line represents no effect in SMD and the broken vertical line indicates the combined (overall) estimates while the diamond stands for the overall pooled SMD effect.

However, one study plot found lying on the right side of the no-effect line had high vitamin D levels in TB patients. The heterogeneity chi-square test was performed to determine the presence of variation between the primary studies that were included in the meta-analysis, and a high variation was noted, (241.28, d.f = 8, (p = < 0.001). The level of variation was then quantified using the $I^2$ test, ($I^2$ = 96.7%). The test overall effect was significant at a, Z = 7.71. (p = < 0.001).

## High circulatory LL-37 levels and low local expression in TB patients

Regarding LL-37 levels across the 12 studies, we found data on either circulatory or local LL-37 expression. Two studies determined LL-37 mRNA expression in cultured samples treated with vitamin D among TB patients. Quantification using real-time PCR of mRNA expressed by macrophages in culture samples from pulmonary TB patients (PTB) and normal health subjects(NHS) stimulated with *Mtb* and treated with 1,25(OH)D3 gave 200 fold change in the TB patients and 10 fold change in NHS, (analytical range 0 -3000change in the fold). CAMP mRNA expression was lower in the NHS compared to the PTB culture supernatants [44]. The second study measured LL-37 mRNA in PBMC cultured samples of TB and non- TB and cells acquired from the local site of infection. Results from this study showed a 4 fold change among the TB Patients and 0.5 fold change in their control group (0.01–1000) [37]. These two studies reported a significant increase in LL-37 mRNA expression in TB patients compared to their controls. In another scenario, two studies determined local LL-37 mRNA expression, one examined lung tissue biopsies and reported a 5 fold change of LL-37

expression in TB patients compared to 9 fold change among their controls (0–15) [43]. Another study examined tissue cells from lung TB lesions and found 0.5 fold change in TB patients and 0.4 fold in their non TB controls [15]. This same study compared PMBC cells from the same patients and controls and found a significant increase in LL-37 mRNA in TB patients compared to the non TB ($p < 0.001$). However, when LL-37 mRNA expression was compared between TB lesion cells and the PBMCs, the difference in expression was remarkable. Eight studies measured circulatory LL-37 levels in both serum and plasma samples. Of these, six studies reported higher circulatory LL-37 levels in TB patients compared to the non-TB individuals [5,11,29,33,39,41]. Two studies found lower serum LL-37 levels in TB patients compared to non -TB individuals [38,42].

## Meta-analysis of LL-37 levels in TB

Only six studies were eligible and these were included in the meta-analysis of LL-37. Fig 3 shows a forest plot of studies that were included in the meta-analysis with details of SMD pooled size estimate of LL-37 levels(SMD = 0.709, CI = (0.496–0.923, p = < 0.001) using the

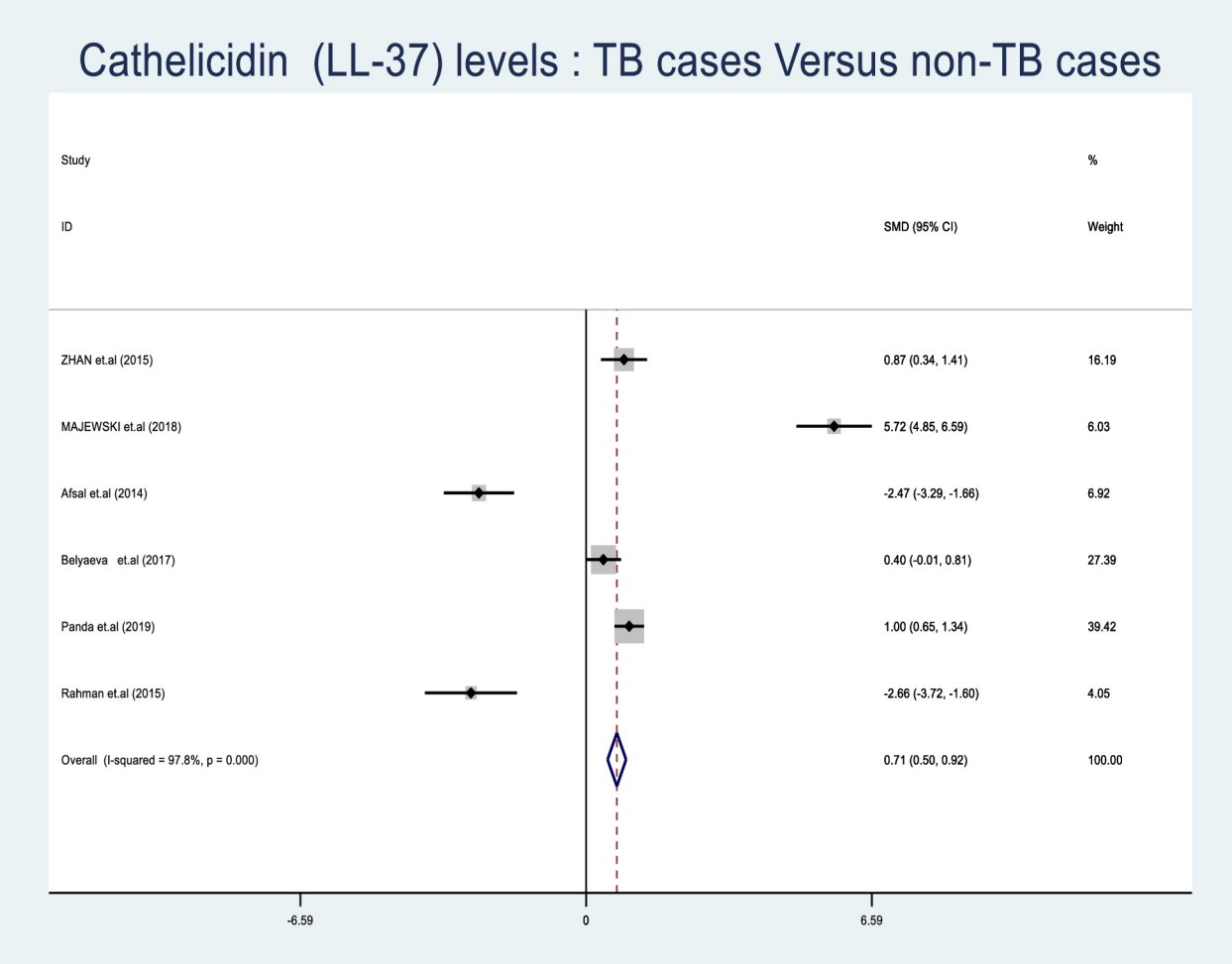

**Fig 3. Forest plot showing the comparison of LL-37 levels among TB patients and non TB individuals: Overall effect of continuous variables using the random effect model.** The solid vertical line represents no effect in SMD and the broken vertical line indicates the combined (overall) estimates while the diamond stands for the pooled SMD effect.

random effect. According to the forest plot, higher LL-37 levels favour the TB patients as shown by the diamond on the right side of the no-effect line, with a significant difference of p < 0.001.Nonetheless, two studies with low LL-37 levels were found on the left side of the no-effect line as already mentioned above. The heterogeneity chi-squared test was to detect for any variation between studies, (230.48, d.f = 5, (p<0.001), and the levels of variation were determined by $I^2$ test,97.8%. Test for overall effect was Z = 6.50p = < 0.001.

## Association of vitamin D status and LL-37 levels in TB patients

Out of the 12 studies reviewed only three performed a linear correlation analysis of vitamin D and LL-37 levels. One study registered a negative correlation between the two molecules; r = -0.10, p = 0.656andrho-0.189, p = 0.248 and the others found no correlation rho = 0.226, (P = 0.082) [15,33,39]. Another study determined the relationship between vitamin D deficiency with both normal and abnormal levels of LL-37 using the chi-square test and found no significant association $X^2$ = 1.31 (df = 1), p = 0.25 [42]. One study reported no correlation between vitamin D and LL-37 molecules. They found no difference in vitamin D levels among TB patients and their controls however higher LL-37 levels were found in the TB patients compared to their control group [41]. Six studies observed that induction of LL-37 expression was determined by vitamin D status and that deficient and insufficient levels would cause a poor response in TB disease. A meta-analysis to determine the pooled effect size estimate of vitamin D and LL-37 levels correlation was not possible in consideration that only three primary studies reported the correlation coefficient [15,33,39].

## Publication bias

Fig 4 shows the funnel plot constructed from vitamin D levels of eight studies analyzed. We found an asymmetrical funnel plot in which five of the studies were on the left lower side, one study on the lower right side, one in the middle left, and one study on the right middle side of

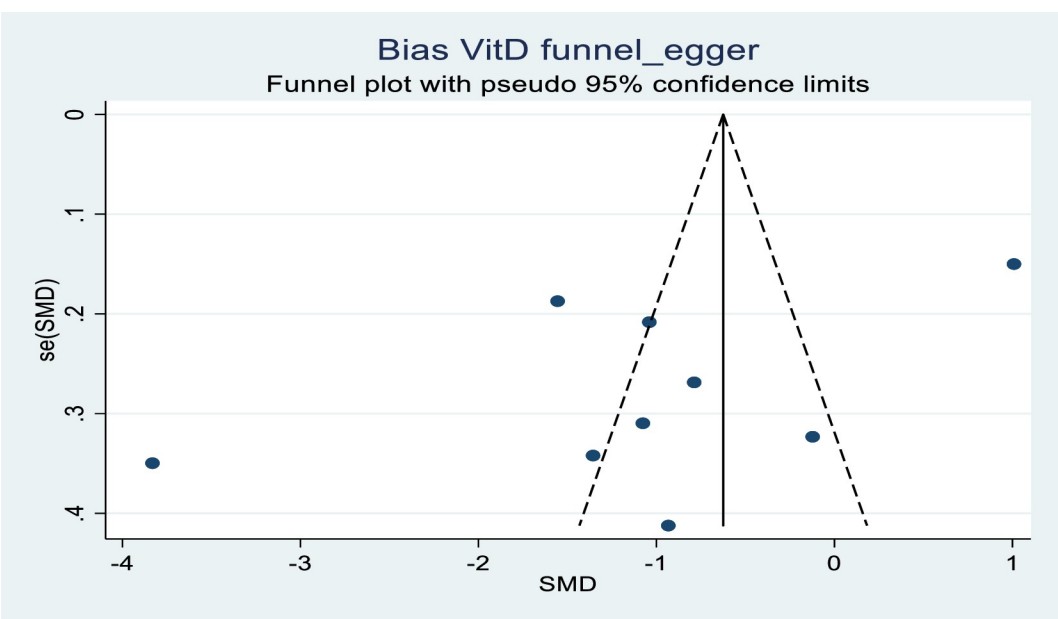

**Fig 4. Funnel plot of standard mean difference of vitamin D levels against standard error of mean difference at CI 95% showing asymmetry in eight highly scattered studies due to publication bias.**

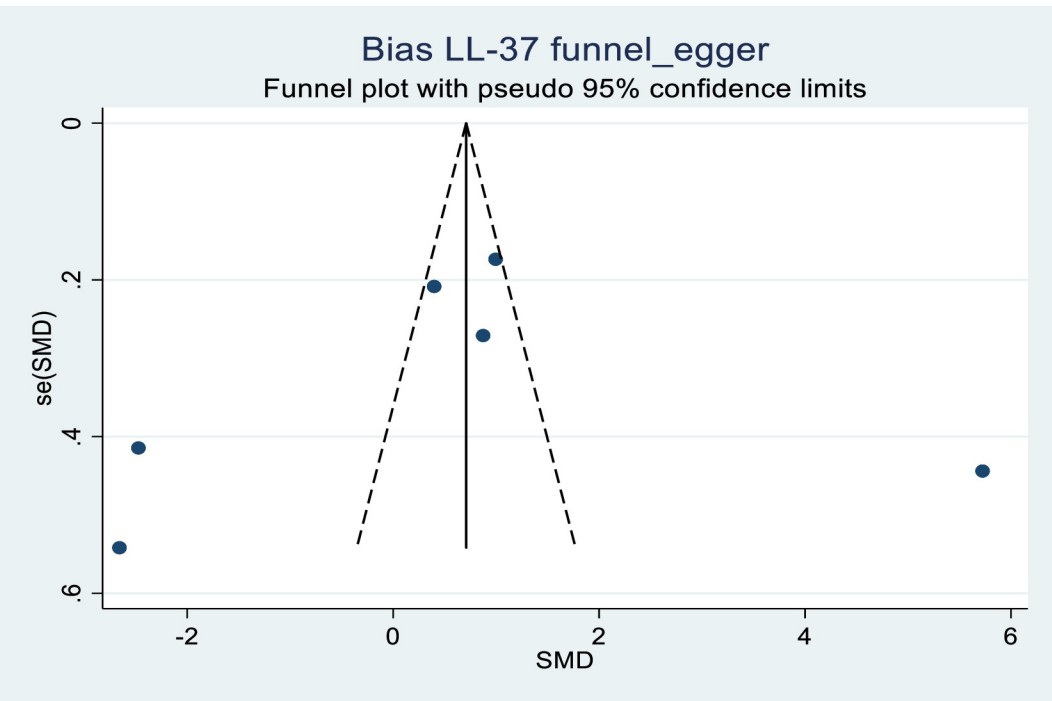

**Fig 5. Funnel plot of standard mean difference of LL-37 levels against standard error of mean difference at CI 95%, in six highly scattered studies due to publication bias.**

the funnel. No study was found at the top of the funnel and two studies were found extremely away from the center of the funnel. Similarly, Fig 5 shows a funnel plot of SMD and SE of LL-37 levels where an asymmetrical funnel plot was constructed by six primary studies. Two study plots were found in the middle left and right of the funnel, one study plot was found on the right top part and three study plots were found outside of the funnel plot. Of the study plots outside the funnel, two were lying on the left far corner way from the center while one was found on the far right. No study plot was found at the inside bottom of the funnel, therefore, presenting an asymmetrical funnel plot.

Concerning the asymmetry of the funnel plots, we further quantified publication bias using two methods for small-study effects were performed. The Begg's rank, correlation test was performed between vitamin D levels SMD and SE at CI 95% in nine studies. The Kendall's score = -6, Std. Dev. of Score = 9.59, Z = 0.52, and (p = 0.602).The Egger's regression test in which effect size was regressed against SE at CI 95% in nine studies, showed an estimated intercept of -9.222,Z = 0.255, SE = 2.51with no significance noted (p = 0.106). The same methods were used to determine publication bias in LL-37 levels SMD and SE at CI 95% in six studies. Begg's test; Kendall's score -3, Std. Dev. of Score = 5.32, Z = 0.38 (p = 0.707). Egger's test = -2.096329 (p = 0.807). Z = SE = 2.15. Both did not show a significant association.

## Discussion

To the best of our knowledge, this might be the first systematic review and meta-analysis to examine the relationship between vitamin D and LL-37 levels. Our findings show that there is a paucity of pooled data on the relationship between vitamin D and LL-37 levels in TB disease. Although enormous research has been done and suggested a strong relationship between vitamin D and TB disease, few studies have explored its role in antimicrobial peptide regulation.

In Africa, we found only one study from Ethiopia, and the majority were from Asia. Based on 12 studies our meta-analysis presents evidence that vitamin D levels are lower in TB patients compared to their controls, and that VDD is associated with high circulatory LL-37 levels and low local LL-37 expression. These results are comparable to previous systematic reviews that found low vitamin D levels in TB patients compared to their controls and that VDD may be a risk factor for TB disease progression [4,6,13,16,45–48]. We were able to demonstrate that patients with TB have high circulatory levels of LL-37. We found that the high circulatory LL-37 levels may be comparable to the up-regulation of LL-37mRNA in the presence of *Mtb* and 1, 25(OH) D3 reported in the two in-vitro studies. Similarly, the higher LL-37 mRNA expression in peripheral PBMCs compared to the low expression in the TB lesion cells is similar to the higher circulatory LL-37 levels found in TB patients. The lower expression in TB lesions is comparable to the low local LL-37 expression found in the lung biopsy. Conversely, some studies have reported higher LL-37 levels in latent TB patients compared to healthy controls, yet others found no difference, however, latent TB patients had lower levels than active TB patients [5,11,49]. This confirms a report from a previous study that the association of vitamin D status with TB disease may be a consequence of enhanced LL-37 production and macrophage activation [46]. Elevated circulatory LL-37 levels may be a result of an increase in bacterial load during the multiplication of *Mtb* and high tissue damage among the active TB disease [34]. In some study observations, elevated LL-37 levels were reported among adults with other bacterial lung diseases like pneumonia and when levels were compared between TB disease and pneumonia the former had higher levels than the latter [30,34]. During the advanced progression of TB infection, the immunomodulatory properties of LL-37 modify macrophage response by modulation of pro-inflammatory and anti-inflammatory cytokine expression hence preserving the host cells. LL-37 is therefore important in innate immunity.

When we examined the link between TB, vitamin D, and LL-37 generally, we found lower vitamin D levels and high LL-37 levels among the TB patients. According to Bhan *et al* [50], a positive correlation between the two molecules was noted when vitamin D levels were < 32 ng/ml. None of the primary studies had sufficient vitamin D among TB patients and non-TB individuals. However alternative studies had a different account, for example in the situation when low vitamin D and low LL-37 levels were found in TB patients consequently could be an event of low bacterial load at the early stage of the disease process. It was important to avoid inconclusive results therefore a meta-analysis on the correlation of vitamin D and LL-37 was not possible on account of missing data on the correlation in primary studies.

About five databases were searched and among the limitations of our systematic review was the shortage of adequate studies on the relationship between vitamin D and LL-37 among TB patients and non TB individuals. Out of twelve studies included nine were included in the meta-analysis of vitamin D levels and only six studies were eligible for the meta-analysis of LL-37 owing to missing data. We detected a high variation between studies shown by the heterogeneity calculations performed. This may be attributed to methodological differences in sample size, study design, and laboratory methods used. Some studies had a very small sample size to detect an effect. Moreover, variable methods were used inthe measurement of vitamin D and LL-37 levels, therefore, different detection limits and units were used. Besides some studies did not present cut-off ranges and therefore did not define vitamin D status. Furthermore, the type of sample that was collected determined the method that was applied for LL-37 estimation. The culture method that determined local LL-37 expression used lung biopsies, culture supernatants of PBMCS and TB lesion tissue. Plasma and serum samples were used to determine LL-37 circulatory levels. In the analysis of the association between the two molecules, only 3 studies reported a correlation, there was missing data in the rest of the studies which made it difficult to perform a meta-analysis.

Funnel plots show a visual relationship between the effect size and precision. Although we found asymmetry in the funnel plots in our study we did not find publication bias in our review. There are several probable causes and these may include selection bias, poor methodological design, heterogeneity, language bias, publication bias, and a few more. We detected high heterogeneity in the vitamin D and LL-37 levels as presented in the results. Effectively we used the random effect size model in the meta-analysis to balance effect size weights in the primary studies. This appropriately gave the quality of our results. Additionally, there could have been selection bias in the control group as seen by the different populations that were recruited in the control group. These ranged from hospital staff, University staff, and students, to the general population these could be the cause of asymmetry. Confirmation of no publication bias was shown by the Begg's and Egger's tests when no significant association was noted.

## Conclusion

Ultimately our pooled study analyses revealed a significant difference in vitamin D and LL-37 levels among TB patients and non-TB individuals. Therefore active pulmonary TB disease is associated with hypovitaminosis D and elevated circulatory LL-37 levels. However low local LL-37 levels were found in TB patients compared to the non- TB individuals. Even so, we did not have sufficient eligible studies to perform a correlation meta-analysis of vitamin D and LL-37 levels.

## Supporting information

**S1 File. PRISMA checklist of study.**
(PDF)

**S2 File. Study protocol of systematic review.**
(PDF)

## Acknowledgments

We are also grateful to the African Centre–MAKCHS for their suggestions in improving methodology.

## Author Contributions

**Conceptualization:** Ester Lilian Acen.

**Data curation:** Ester Lilian Acen.

**Formal analysis:** Ester Lilian Acen, Joseph Musaazi.

**Investigation:** Ester Lilian Acen, Bill Nkeeto.

**Methodology:** Ester Lilian Acen, Bill Nkeeto, David Patrick Kateete.

**Software:** Joseph Musaazi.

**Supervision:** Irene Andia Biraro, William Worodria, Moses L. Joloba, David Patrick Kateete.

**Writing – original draft:** Ester Lilian Acen, Irene Andia Biraro, William Worodria, Moses L. Joloba, David Patrick Kateete.

**Writing – review & editing:** Ester Lilian Acen, Irene Andia Biraro, William Worodria, Moses L. Joloba, David Patrick Kateete.

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
