## [Decision Letter · Decision Letter 0]

18 Mar 2021

PONE-D-20-24487

Impact of vitamin D status and cathelicidin antimicrobial peptide on adults with active pulmonary TB globally: A systematic review and meta-analysis

PLOS ONE

Dear Dr. ACEN,

Thank you for submitting your manuscript to PLOS ONE. After careful consideration, we feel that it has merit but does not fully meet PLOS ONE’s publication criteria as it currently stands. Therefore, we invite you to submit a revised version of the manuscript that addresses the points raised during the review process.

We look forward to receiving your revised manuscript.

Kind regards,

Jennifer A. Hirst, DPhil

Academic Editor

PLOS ONE

Journal Requirements:

3. Please address the following:

- Please update the last search to allow the inclusion of studies published in the past 12 months.

- Please ensure you have included the full electronic search strategy for at least one database and uploaded it as an additional file.

Additional Editor Comments:

In addition to the comments there are a few other points that this manuscript needs before publication:

Please avoid abbreviations in the abstract (TB and LL-37)

Data should be plural (page 7 Data management and statistical analysis second line should read data from final studies were….) – please check through rest of manuscript

Please give justification for pooling SMD. Were Vitamin D and LL-37 measured using different units?

Results

Table 1:

• please reformat in landscape

• Define Nc, Please present as NC or Nc, currently both used

• Define PTB, PBMC

Reviewers' comments:

Reviewer's Responses to Questions

**Comments to the Author**

1. Is the manuscript technically sound, and do the data support the conclusions?

Reviewer #1: Yes

Reviewer #2: Yes

2. Has the statistical analysis been performed appropriately and rigorously? 

Reviewer #1: Yes

Reviewer #2: Yes

3. Have the authors made all data underlying the findings in their manuscript fully available?

Reviewer #1: Yes

Reviewer #2: Yes

4. Is the manuscript presented in an intelligible fashion and written in standard English?

Reviewer #1: Yes

Reviewer #2: Yes

5. Review Comments to the Author

Reviewer #1: This study of this systematic review and meta-analysis was carried out with rigor and achieved the objectives it set out to.

Although the existing studies have not allowed to find more consistent results, the systematic review and meta-analysis performed by the authors correctly follows the steps for carrying out these studies. Furthermore is relevant for clinical and biological areas.

Protocol was previously registered.

Participants, exposure, comparator and outcome (PECO) are clear. Also, eligibility criteria, quality of the studies, information sources, search strategy, data collection, risk of bias are also well explained in methods.

In what concerns meta-analysis, size variability, heterogeneity, methods and interpretation of results are all clear in the article.

However, minor reviews are suggested:

Aims

The title “Impact of vitamin D status and cathelicidinantimicrobial peptide on adults with active pulmonary TB globally: A systematic review and meta-analysis” does not reflect the objective regarding the relationship between Vitamin D status and cathelicidin levels on adults with active pulmonar TB.

(although it is the title of the protocol registered in PROSPERO)

Protocol aims: “The aim of this review is to determine the association of vitamin D status with cathelicidinexpression among adults with pulmonary disease.

Article Pag2. “The systematic review attempts to define the relationship between cathelicidin levels and vitamin D status in TB disease”

Article Pag4. “The objective of our systematic review was to compare vitamin D and LL-37 levels among TB patients to non-pulmonary TB (…) and to determine the association between vitamin D and LL-37 and any ….”

This last aim is clear and reflects the performed study so would suggest adjusting the previous ones.

Results:

There are two different types of studies, one is the determination of LL-37 plasma levels and the other is the cell expression of LL-37. It could be more explicit in the text. Also, for Vitamin D.

Pag 10. Table 1. Would suggest a reference in author column. It will help the readers find the article.

Pag 12. “The lowest vitamin D levels of >10ng/ml (…) six studies reported vitamin D deficiency levels of >20ng/ml”. This statement is not clear.

References

Suggest to review the references. Some are not complete (for example reference 8)

Reviewer #2: This study investigated the impact of vitamin D and LL-37 level on TB. The found that a significant difference was observed in both vitamin D and cathelicidin levels among TB patients and non-TB individuals (p= < 0.01). Overall, the study is interesting and the manuscript is well-designed. I just have several suggestions.

1. Please clarify which is the target of this study – TB or active pulmonary TB.

2. Please add some data in the abstract’s result section.

3. The introduction is too long.

4. Please add the level of Vit D and LL-37 and case number in TB and non-TB in the table 1.

6. PLOS authors have the option to publish the peer review history of their article (what does this mean?). If published, this will include your full peer review and any attached files.

Reviewer #1: No

Reviewer #2: No

---

## [Editor Report · Decision Letter 1]

3 May 2021

PONE-D-20-24487R1

Impact of vitamin D status and cathelicidin antimicrobial peptide on adults with active pulmonary TB globally: A systematic review and meta-analysis

PLOS ONE

Dear Dr. ACEN,

Thank you for submitting your manuscript to PLOS ONE. After careful consideration, we feel that it has merit but does not fully meet PLOS ONE’s publication criteria as it currently stands. Therefore, we invite you to submit a revised version of the manuscript that addresses the points raised during the review process.

Please clearly highlight the changes that were made to the manuscript to help the reviewers and Editor.

We look forward to receiving your revised manuscript.

Kind regards,

Jennifer A. Hirst, DPhil

Academic Editor

PLOS ONE

Journal Requirements:

Additional Editor Comments (if provided):

Unfortunately I am unable to clearly see the changes you have made as requested by the reviewers. Please upload a revised version of the manuscript with the changes highlighted so it can be confirmed that the changes have been made.

Many thanks

---

## [Editor Report · Decision Letter 2]

24 May 2021

Impact of vitamin D status and cathelicidin antimicrobial peptide on adults with active pulmonary TB globally: A systematic review and meta-analysis

PONE-D-20-24487R2

Dear Dr. ACEN,

We’re pleased to inform you that your manuscript has been judged scientifically suitable for publication and will be formally accepted for publication once it meets all outstanding technical requirements.

Kind regards,

Jennifer A. Hirst, DPhil

Academic Editor

PLOS ONE
---

## [Editor Report · Acceptance letter]

3 Jun 2021

PONE-D-20-24487R2 

Impact of vitamin D status and cathelicidin antimicrobial peptide on adults with active pulmonary TB globally: A systematic review and meta-analysis 

Dear Dr. Acen:

I'm pleased to inform you that your manuscript has been deemed suitable for publication in PLOS ONE. Congratulations! Your manuscript is now with our production department. 

Kind regards, 

on behalf of

Dr. Jennifer A. Hirst 

Academic Editor

PLOS ONE